# Graph Supervised Contrastive Learning for Geodemographics

## Abstract

Geodemographic analysis is essential for understanding population characteristics and addressing socio-economic disparities across regions. However, limited research has been conducted on modelling changes in demographic data over time using Graph Neural Networks (GNNs). In this study, we address this gap by leveraging GNNs to model correlations between the 2011 census data (England & Wales), observing changes over time, and the Output Area Classification 2021, which reflects socio-economic differences between Output Areas. We propose a novel framework that utilises Supervised Contrastive Learning on graphs to obtain robust OA embeddings, with a particular focus on improving the model's performance for minority classes. To evaluate the effectiveness of our framework, we conducted two downstream tasks based on the 2021 OA embeddings. Our results demonstrate that the proposed approach provides valuable insights for geodemographic analysis and offers policymakers a useful tool for assessing socio-economic transitions over time, and planning ahead on the basis of it.

## 1 Introduction

Demographics is the study of the statistical characteristics of populations, including factors such as age, gender, and income. These data are often utilised to understand population trends, forecast changes Mielczarek & Zabawa (2021), and guide policy decisions. **Geodemographics** extends demographic analysis by integrating geographical and sociological data, enabling the study of populations within specific locations.

Geodemographic analysis is essential for understanding the spatial distribution of population characteristics, providing a detailed view of socio-economic diversity and its geographic implications (Birkin & Clarke, 2009; Singleton & Longley, 2009; Webber, 2007). By integrating demographic data with geographic information, geodemographics uncovers patterns related to inequality, access to services, and consumer behaviour in specific locations and across locations. These insights enable policymakers and businesses to make informed, location-based decisions—whether for public health initiatives (Abbas et al., 2009; Guan, 2020; Brown et al., 1991), urban planning (Batey, 2022), or market segmentation (Farr et al., 2008). The ability to analyse data at multiple geographic scales, from neighbourhoods to regions, enhances the effectiveness of targeting interventions and allocating resources to areas with diverse socio-economic profiles. Overall, spatial analysis through geodemographics is crucial for addressing regional disparities and improving service delivery by tailoring resources to meet the specific needs of different communities.

Geodemographic data fundamentally consists of two primary components: the spatial relationships between geographic areas and the demographic characteristics of those areas. Spatial relationships refer to the connections and proximities between locations, such as neighbourhoods or regions, while demographic characteristics encompass variables including income levels, age distributions, educational attainment, and housing attributes. Traditionally, the analysis of geodemographic data using computational models necessitates a significant preprocessing phase. This phase often includes extensive feature engineering, where spatial relationships are carefully integrated into the attribute data to enable models to capture the intricate correlations between geographic and demographic factors effectively (De Sabbata & Liu, 2023; Wyszomierski et al., 2024). Although essential for developing accurate and predictive models, this process is time-intensive and requires substantial expertise in both data science and domain-specific knowledge.

The emergence of Graph Neural Networks (GNNs) has introduced a transformative paradigm in geodemographic studies, providing a novel methodological approach (De Sabbata & Liu, 2023). GNNs are specifically designed to process structured data in an end-to-end manner, reducing the need for extensive manual feature engineering. Since geodemographic data can naturally be represented as graphs—capturing the relationships between nodes, such as the connections between geographic areas—GNNs allow for the development of more robust models capable of addressing complex social challenges. This approach enables the detection of patterns that were previously difficult to discern using traditional methods, representing a significant advancement in the field and enabling more sophisticated and precise analyses of geodemographic data.

In this research, we construct graphs using datasets from the 2011 and 2021 Censuses of England and Wales (ONS, 2011), along with geospatial data for Output Areas (OAs) from the same years. An OA is the smallest census geographic unit, each comprising 40 to 250 households (ONS, 2011). Each node in the graph represents an OA, with adjacency relationships determined using Queen's contiguity (Gwani & Sek, 2024). Node attributes are derived from census data, covering various sectors such as employment, residence, living arrangements, ethnicity, and origins. These attributes are aligned with the data selection criteria of the 2021 UK Output Area Classification (OAC) (Wyszomierski et al., 2024), produced through a collaboration between the Office for National Statistics and University College London.

As mentioned earlier, a key objective of geodemographics is to identify changes and observe regional differences. However, limited research has focused on utilising GNNs to model the relationship between changes in census data and regional differences, while incorporating geographic information. In this study, we address this gap by using GNNs to model the correlations between the 2011 census data, observed changes over time, and the OAC, which reflects the differences between OAs in 2021.

However, GNNs often face two critical issues: oversmoothing (Rusch et al., 2023) and oversquashing (Nguyen et al., 2023). These issues can degrade learning performance and lead to indistinguishable embeddings. Oversmoothing is especially problematic in imbalanced datasets (Yang et al., 2023), where nodes from minority classes are overwhelmed by the majority, resulting in a loss of important information. In OAC 2011 and 2021, the minority classes represent more deprived groups (e.g., Hard-Pressed Living and Legacy Communities), and it is crucial that they are not ignored.

As noted in (Nguyen et al., 2023), graphs with positive curvature bring nodes closer together, accelerating the mixing of information. This can lead to oversmoothing, where node representations become overly similar, preventing the model from distinguishing between different nodes. Conversely, graphs with negative curvature or stretched structures tend to distribute information unevenly, leading to oversquashing. In this case, information from distant nodes is compressed into a few intermediary nodes, limiting the model's ability to represent all relationships. Figure 1 illustrates the curvature of edges in the UK graph, highlighting its complex structure and the potential for oversmoothing and oversquashing during GNN training.

To address these challenges, we propose a framework using Supervised Contrastive Learning (SCL) (Khosla et al., 2020) on graphs to obtain embeddings for OAs. The 2011 OA embeddings are learned through SCL, using 2011 OA graph as input and the 2011 OAC (Gale et al., 2016) as labels. The 2021 OA embeddings are then generated by incorporating both the 2011 OA embeddings and changes in OA attributes between 2011 and 2021, with the 2021 OAC as labels. We hypothesise that the 2011 OA embeddings implicitly capture the socio-economic conditions of the OAs at that time, providing valuable insights into their status. By combining the 2011 OA embeddings with the changes in OA attributes over the decade, we aim to infer the states of OAs in 2021, where the new states are represented by the corresponding labels. While SCL was originally designed for image classification tasks, we extend its application to graph data. Our motivation for using SCL is to improve the model's performance for minority classes by mapping those embeddings into a more distinguishable feature space. By pulling together embeddings of the same classes while pushing apart those of different classes, SCL helps ensure minority class embeddings are not overwhelmed by majority classes.

Moreover, we conducted both supervised and unsupervised classification tasks using the 2021 OA embeddings to demonstrate the advantages of our proposed framework for geodemographic analysis. First, we trained and tested a fully-connected neural network using these embeddings to predict the 2021 labels (Wyszomierski et al., 2024). This approach provides policymakers and stakeholders

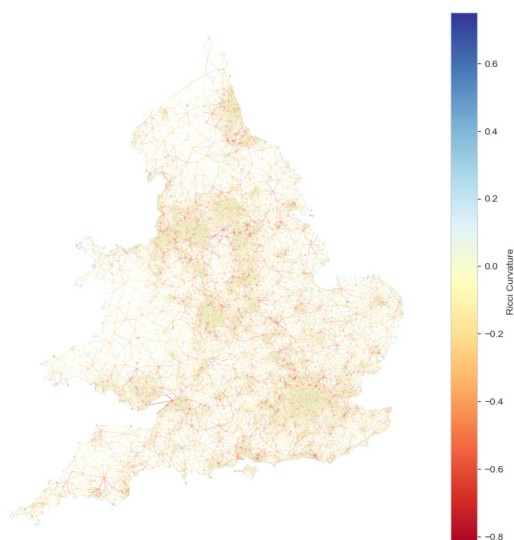

Figure 1: The average curvatures of the UK graph.

with a valuable tool for decision-making, allowing them to review and simulate changes between 2011 and 2021. If certain areas are classified as deprived in 2021, the tool allows for adjustments to the variables representing changes, potentially illustrating how these areas could transition to less deprived groups. Second, inspired by the work of De Sabbata & Liu (2023), we utilised Bayesian Gaussian Mixture Models (BGMM) based on the embeddings generated by SCL to re-classify the OAs.

The rest of this paper is organised as follows. In Section 2, we review related work in the fields of graph learning and clustering algorithms applied to geodemographics. Section 3 introduces our proposed framework, focusing on using graph SCL to generate OA 2021 embeddings from the 2011 OA graph data and changes in OA attributes between 2011 and 2021. In Section 4, we present experiments aimed at predicting the 2021 OAC using these embeddings. Section 5 explores the application of clustering techniques on the 2021 OA embeddings to uncover regional patterns and socio-economic groupings. Finally, Section 6 concludes the paper with a summary of our findings and potential future directions for this line of research.

## 2 RELATED WORK

### 2.1 GRAPH NEURAL NETWORKS

As research into deep learning methods for graph data continues to expand, more and more GNN solutions are being developed (Wu et al., 2020). Driven by advances in convolutional neural networks(CNNs) (LeCun et al., 1995) and variational autoencoders(VAE) (Kingma & Welling, 2013), researchers have created new methods for summarizing and manipulating graph data. Currently, common GNN solutions include GCN (Kipf & Welling, 2016a), VGAE (Kipf & Welling, 2016b) inspired by VAE, Graph Transformer (Shi et al., 2020) which inspired by the Transformer model (Vaswani, 2017), and GraphSAGE (Hamilton et al., 2017).

Kipf & Welling (2016a) present a scalable approach for semi-supervised learning on graph-structured data using an efficient variant of convolutional neural networks that operate directly on graphs. This model scales linearly in the number of graph edges, and the learned hidden layer representations encode both local graph structure and node features. Variational Graph Auto-Encoders (VGAEs) represent a robust methodology in graph representation learning by integrating variational inference with GNNs to derive latent representations useful for node classification, link prediction, and graph generation. First introduced by Kipf & Welling (2016b), VGAEs employ a Graph Convolutional Network (GCN) for the encoder, facilitating efficient unsupervised learning with graph-structured data. Concurrently, GraphSAGE Hamilton et al. (2017), another pivotal development in

graph representation learning, specialises in creating node embeddings for expansive and intricate graphs. GraphSAGE's innovation is its use of inductive learning, which, unlike transductive methods that learn specific embeddings for each node, learns a function to produce embeddings based on the attributes of nodes and the characteristics of their surrounding local network.

To address the challenges of combining feature and label propagation, the proposed Unified Message Passing Model (UniMP) Shi et al. (2020) introduces a Graph Transformer network that takes both node features and label embeddings as input, unifying feature and label propagation within a single framework. To prevent overfitting from self-loop label information, UniMP incorporates a masked label prediction strategy, where a portion of the input labels is randomly masked, and the model is trained to predict these masked labels. By effectively combining the complementary strengths of GNNs and Label Propagation Algorithms (LPAs), UniMP achieves state-of-the-art results on semi-supervised classification benchmarks in the Open Graph Benchmark.

## 2.2 CLUSTERING ALGORITHMS

While it is useful for policy makers to classify areas according to features, there is no observable objective ground truth classification and feature data is high-dimensional, reflecting the complexities present in society. In the UK, researchers have recently preferred classifying *Output Areas (OA's)* using the well-known K-Means algorithm (Gale, 2014; Gale et al., 2016; De Sabbata & Liu, 2023; Wyszomierski et al., 2024). K-Means is straight-forward to use, efficient, works with hierarchical clustering, and produces understandable outputs (Wyszomierski et al., 2024); all important factors when creating policy from results.

Some issues have been identified with K-Means clustering when applied to geodemographic data. Vickers & Rees (2011) found that manual classification by people in OAs did not match the K-means clustering with mean accuracy across classes of 0.839 with standard deviation 0.095. Potential issues relate to the assumptions of K-means (MacQueen, 1967): Firstly, it is not clear the feature space forms approximate hyperspherical clusters in Euclidean space, as required for accurate K-means clustering of high-dimensional data. Secondly, it is uncertain clusters are separable into distinct high-dimensional convex Voronoi cells which reflect a diverse UK population.

To solve some of these issues, De Sabbata & Liu (2023) use an unsupervised *Variational Graph Autoencoder (VGAE)* to generate embeddings from features, forcing them onto a latent-space which approximates a hypersphere, before clustering with K-Means. By using a GNN, this method also ensures latent representations include the effect of geographic relationships between OAs. A different approach to clustering was taken by Muscatiello (2023), who used *Bayesian Profile Regression (BPR)* with respect to incidence of respiratory issues. Where a target variable is unknown, *Bayesian Gaussian Mixture Models (BGMM)* (Roberts et al., 1998), with full *covariance matrix* and *Dirichlet Process with stick-breaking* (Görür & Edward Rasmussen, 2010; Begehr & Panfilov, 2022), supports non-hypersphere clusters and does not require them to be linearly separable, overcoming some of the identified issues (Chang et al., 2020).

## 3 THE FRAMEWORK

Figure 2 illustrates the proposed framework. First, the OA 2011 graph is processed using SCL with OAC 2011 as the labels. During this process, three types of nodes are involved: anchor, positive, and negative. The anchor node represents a sample node whose embedding is used as a reference. Positive nodes belong to the same class as the anchor, while negative nodes are from different classes.

The loss function of SCL is defined as

$$\mathcal{L}_{\text{SCL}} = \sum_{v \in V} \mathcal{L}_{\text{SCL},v} = \sum_{v \in V} \frac{-1}{|P(v)|} \sum_{p \in P(v)} \log \frac{\exp\left(\boldsymbol{h}_v \cdot \boldsymbol{h}_p / \tau\right)}{\sum_{n \in N(v)} \exp\left(\boldsymbol{h}_v \cdot \boldsymbol{h}_n / \tau\right)}, \tag{1}$$

where $\mathcal{L}_{\text{SCL}}$ denotes the total supervised contrastive loss, which is computed as the sum of individual losses over all nodes $v$ in the set $V$ (the graph's node set). For each node $v$, $\mathcal{L}_{\text{SCL},v}$ is the supervised contrastive loss for that node, computed by comparing its embedding $\boldsymbol{h}_v$ with both positive and negative samples. $P(v)$ refers to the set of positive nodes for $v$, i.e., nodes that belong to the same

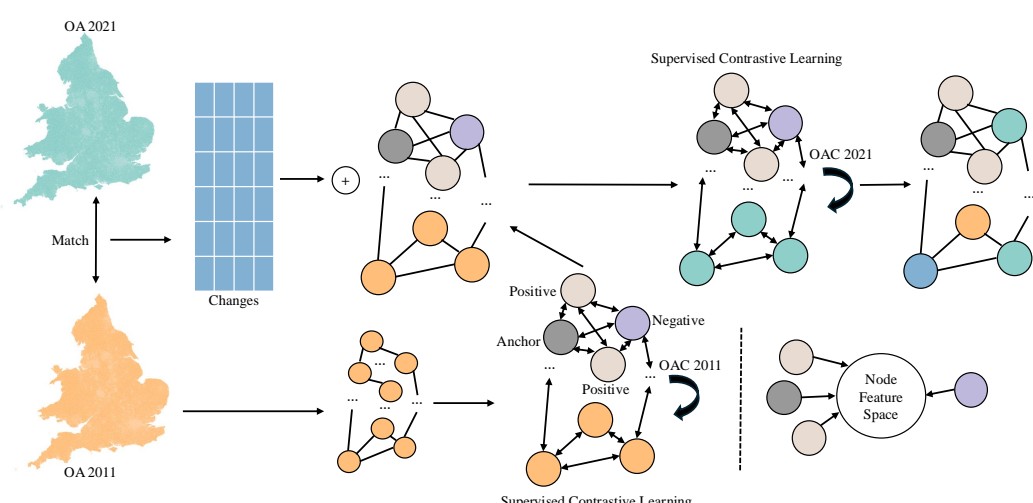

Figure 2: The proposed framework for geodemographic graph analysis utilises the OA 2011 graph and the changes in node attributes between 2011 and 2021 as input data. The output of this framework is the embeddings of OA 2021. The Output Area Classification (OAC) is employed for supervised contrastive learning, serving as the label for guiding the learning process.

class as $v$. The inner summation over $P(v)$ computes the similarity between $v$'s embedding and the embeddings of these positive nodes. The **numerator** of the logarithmic term represents the similarity score between the anchor node $v$ and a positive node $p$, computed using the dot product of their embeddings (i.e., $\boldsymbol{h}_v \cdot \boldsymbol{h}_p$) scaled by a temperature parameter $\tau$. The dot product serves as a measure of similarity between two node embeddings. The **denominator** sums the similarities between node $v$ and all nodes in $N(v)$, where $N(v)$ is the set of all nodes (both positive and negative samples) considered for contrastive learning. The negative samples are represented by nodes from different classes than $v$, and the goal is to minimise their contribution to the similarity score. By minimising this loss function, the model encourages node embeddings of the same class to be similar (high dot product) and node embeddings of different classes to be dissimilar (low dot product), resulting in more discriminative node embeddings for downstream tasks such as classification or clustering.

During the SCL process, the GNN model is trained to bring the embeddings of the anchor and positive nodes closer together in the feature space while pushing the negative node embeddings further apart. This process results in more distinguishable and class-specific embeddings, which are crucial for geodemographic classification tasks. Additionally, this approach helps address the issue where a node is geographically surrounded by nodes with different labels. In such cases, the node becomes vulnerable to producing indistinguishable embeddings, as GNNs rely heavily on message passing, which could lead to confusion in the node's representation.

Once the embeddings for OA 2011 are learned, the changes in OA attributes between 2011 and 2021 are concatenated with the 2011 node embeddings. This combined information is then passed through a second round of SCL, with OAC 2021 as the labels, generating updated embeddings for OA 2021.

It is important to note that OAC 2011 and OAC 2021 are not directly aligned. To address this issue, we hypothesise that the 2011 embeddings can capture the underlying state of the OAs. When the changes in OA attributes between 2011 and 2021 are concatenated with the 2011 embeddings, the model should be able to infer the new status of the OAs in 2021. For instance, if nodes have similar 2011 embeddings and exhibit similar changes, their 2021 embeddings should also be similar, leading them to be classified into the same group.

The final embeddings for 2021 OAs are used for downstream tasks such as classification and clustering. These embeddings capture the evolving socio-economic characteristics of regions, as the

framework integrates both historical data (2011) and the changes observed over the decade. Algorithm 1 summaries the workflow of our proposed framework.

---

**Algorithm 1** Geodemographic Framework with Supervised Contrastive Learning

---

1: **Input:** OA 2011 graph $G_{2011}$, OAC 2011 labels $L_{2011}$, OAC 2021 labels $L_{2021}$, Changes in attributes $C_{2011 \to 2021}$
2: **Output:** OA 2021 embeddings $E_{2021}$
3: **Step 1: Initial Embedding for OA 2011**
4: Initialize the OA 2011 embeddings $E_{2011}$ using node attributes and OAC 2011 labels $L_{2011}$
5: Perform Supervised Contrastive Learning on $G_{2011}$ with $L_{2011}$ as labels
6: Obtain the 2011 embeddings $E_{2011}$ after training
7: **Step 2: Incorporate Changes from 2011 to 2021**
8: Concatenate $E_{2011}$ with the changes in attributes $C_{2011 \to 2021}$
9: Form the updated node features for 2021 as $F_{2021} = \text{concat}(E_{2011}, C_{2011 \to 2021})$
10: **Step 3: Embedding for OA 2021**
11: Initialize the OA 2021 embeddings $E_{2021}$ using $F_{2021}$
12: Perform Supervised Contrastive Learning on $G_{2021}$ with $L_{2021}$ as labels
13: Obtain the final OA 2021 embeddings $E_{2021}$ after training
14: **Step 4: Downstream Tasks**
15: Use the final OA 2021 embeddings $E_{2021}$ for classification and clustering tasks

---

## 4 THE PREDICTIONS OF OUTPUT AREA CLASSIFICATION

The census, conducted by the Office for National Statistics (ONS) every ten years, aims to capture the population and household conditions in England and Wales. The survey includes questions on various aspects such as healthcare, household composition, and demographic characteristics, with data summarized across different geographic scales. Previous studies by Gale et al. (2016) and Wyszomierski et al. (2024) classified OAs based on the 2011 and 2021 censuses, respectively [1], and these classifications are used as labels in our experiments. However, the OAC results for 2011 [2] and 2021 [3] differ, as the 2011 OAC covers England, Scotland, Wales, and Northern Ireland, while the 2021 OAC focuses solely on England and Wales. To ensure consistency in our experiments, we limit our analysis to the OAs in England and Wales for both 2011 and 2021.

### 4.1 DATA CLEANING AND PREPROCESSING

Due to differences in the questions and response options between the 2011 and 2021 censuses, some questions were merged, while others had their wording. To ensure data consistency and enable comparison with the OAC results presented by Wyszomierski et al. (2024), all 60 variables from their study were reviewed. Corresponding variables in both the 2011 and 2021 datasets were re-identified. For variables that had changed, alternative questions and responses that closely matched the originals were selected. As a result, a total of 61 variables were generated.

Additionally, OAs serve as the smallest geographical units in the census. Due to population shifts and household changes, there are some differences between the OA divisions in 2011 and 2021. The 2021 OAs consist of a mix of unchanged 2011 OAs and modified 2021 OAs. Therefore, using the reference table[4], the 2011 OAs were merged and split to align with the 2021 OAs. As this merging and splitting process affects the values corresponding to each area, specific strategies were adopted: for merging, the average value of the regions being combined was used, and for splitting, the values of the original region were copied into the new subdivided OAs. This ensured that all data from 2011 and 2021 were fully aligned. Before conducting the experiments, all the necessary data from 2011 and 2021 were aligned based on OAs. We used percentages for the variables, meaning the variables represent the percentage of responses within each OA.

---

[1] https://www.ons.gov.uk/methodology/geography/ukgeographies/statisticalgeographies
[2] https://data.cdrc.ac.uk/dataset/output-area-classification-2011
[3] https://data.cdrc.ac.uk/dataset/output-area-classification-2021
[4] https://github.com/jakubwyszomierski/OAC2021-2/tree/main

Table 1: The comparison of the results from the prediction tasks includes overall accuracy and F1 scores for the 8 classes. Classes 1 to 8 represent the following categories, respectively: Retired Professionals, Suburbanites and Peri-Urbanites, Multicultural and Educated Urbanites, Low-Skilled Migrant and Student Communities, Ethnically Diverse Suburb-an Professionals, Baseline UK, Semi- and Un-Skilled Workforce, Legacy Communities.

|  |  | SCL+NN | | SCL+GNN | | GNN | |
| --- | --- | --- | --- | --- | --- | --- | --- |
|  |  | GT | GraphSage | GT | GraphSage | GT | GraphSage |
|  | Accuracy(%) | 73.27±0.004 | 76.64±0.002 | 82.63±0.002 | 81.39±0.010 | 90.69±0.010 | 90.30±0.009 |
|  | 1 | 0.785±0.006 | 0.810±0.001 | 0.852±0.006 | 0.855±0.006 | 0.897±0.007 | 0.904±0.012 |
|  | 2 | 0.707±0.006 | 0.771±0.002 | 0.835±0.007 | 0.826±0.006 | 0.918±0.008 | 0.916+0.012 |
|  | 3 | 0.844±0.004 | 0.842±0.005 | 0.895±0.002 | 0.871±0.010 | 0.944±0.005 | 0.947+0.003 |
| F1 Score | 4 | 0.819±0.002 | 0.781±0.006 | 0.887±0.002 | 0.875±0.007 | 0.941±0.015 | 0.941+0.010 |
|  | 5 | 0.659±0.005 | 0.685±0.003 | 0.789±0.006 | 0.762±0.017 | 0.913±0.009 | 0.899±0.013 |
|  | 6 | 0.627±0.009 | 0.698±0.003 | 0.773±0.005 | 0.747±0.018 | 0.894±0.014 | 0.889±0.010 |
|  | 7 | 0.765±0.003 | 0.807±0.002 | 0.848±0.005 | 0.830±0.012 | 0.914±0.009 | 0.902±0.006 |
|  | 8 | 0.571±0.007 | 0.645±0.003 | 0.000±0.000 | 0.000±0.000 | 0.000±0.000 | 0.000±0.000 |

## 4.2 EVALUATION

According to the framework shown in Figure 2, the 2011 OA graph was used to train the Graph Transformer and obtain the 2011 OA embeddings. Notably, using the full 2011 dataset for training ensures that the embeddings capture the latent features of the 2011 data without significant information loss, providing a relatively robust starting point for the subsequent work. The size of the 2011 embeddings was set to 61 to maintain alignment with the work of Wyszomierski et al. (2024).

After calculating the changes in OA attributes between the 2011 and 2021 data, these changes were concatenated with the corresponding 2011 OA embeddings. The resulting dataset was then split into training and testing sets at an 8:2 ratio. Fixed data masks were used to generate the two sets and were preserved throughout the experiments to ensure that the test data remained unseen by the model during training. The training data was then used to train both the Graph Transformer and GraphSAGE models, generating the OA 2021 embeddings. This process ensured that no data leakage occurred. Finally, a neural network was trained on the 2021 OA embeddings using the 2021 OAC as labels.

The comparison results are presented in Table 1, with all values derived from five independent runs, reporting both the mean and standard deviations. The table summaries the outcomes of various comparative experiments, including average accuracy and F1 scores for the 8-class classification task. The first column, "SCL+NN" presents our proposed approach, which the 2011 embeddings were initially generated using SCL, followed by the generation of the 2021 embeddings through another round of SCL, after which a neural network was trained on the 2021 embeddings. The second column, "SCL+GNN" shows the results where a graph neural network was directly trained on the 2011 embeddings and the changes in OA attributes between 2011 and 2021 to predict OAC. The third column, "GNN" represents the results of training a graph neural network directly on the original 2011 data along with the change values for OAC classification.

## 4.3 DISCUSSION

The experimental results in Table 1 show that our proposed framework does not achieve the highest overall accuracy. However, considering that the framework has two objectives—OAC classification and improving model performance on minority classes—we believe the performance is acceptable, as the F1-scores for all eight classes are above 0.5. Additionally, GraphSAGE outperforms the Graph Transformer within our framework, while the opposite is observed in the other two experiments. This may be because GraphSAGE is better suited to our framework, and the use of SCL significantly aids in capturing information and improving performance.

Furthermore, it is evident that all models in the other two experiments have an F1 score of 0 for the 8th class, indicating that none of the OAs were classified into the 8th class correctly. This is unfair for the OAs in the 8th class, as the trained GNN models may prioritise higher overall accuracy by neglecting samples from the minority class. Through data exploration, it was found that the 8th category has the smallest amount of data, accounting for only 2.57% and 2.63% of the training and testing datasets, respectively. Although the dataset is imbalanced, such biased predictions can lead to harmful outcomes in geodemographics, as OAs in the 8th class are not treated equally with those in other classes. In contrast, our proposed framework demonstrates different results, with relatively balanced F1 scores across all 8 classes.

## 5 CLUSTERING FOR OUTPUT AREA CLASSIFICATION

Following Wyszomierski et al. (2024), who fit K-Means models with a range of $k = [1, 11]$ clusters, this project fits BGMM models with the same range. As outlined in Section 3, this project uses embeddings for clustering instead of feature data, leading to better separation of records.

BGMM model clusters are examined using a clustergram (Fleischmann, 2023), useful in showing the separation between clusters and how data cluster assignments change with the number of clusters. In this case, Figure 3 confirms the choice of 8 clusters made by Gale (2014) is robust to different cluster methods, but more clearly separates clusters. For comparison, the feature-space K-Means cluster separation is approximately 0.1 between 3 pairs of the 8 clusters (Wyszomierski et al., 2024).

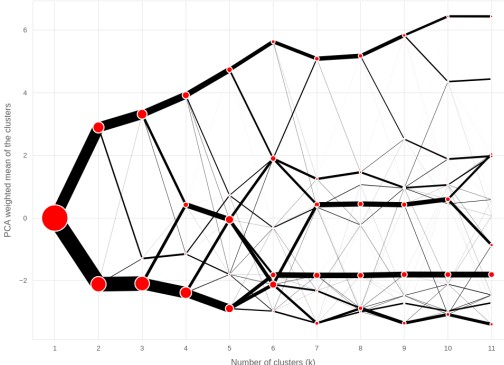

Figure 3: A clustergram of the 2011 OA latent-space embeddings produced by supervised contrastive learning and clustered with BGMM. In line with Wyszomierski et al. (2024), this result suggests 8 clusters is a good choice, but maintains better separation of clusters than seen with K-Means clustering of feature space.

Since K-Means has been a popular choice when clustering for OA classification (Gale, 2014; De Sabbata & Liu, 2023; Wyszomierski et al., 2024), the BGMM clusters were compared to K-Means clusters of the same embeddings. As shown by Figure 4, the coincidence of clusters between BGMM and K-Means calculated using linear sum assignment (Crouse, 2016) and a confusion matrix, shows only 53% similarity between cluster assignments.

Visualising classifications geospatially, with a focus on Greater London and Greater Manchester Combined Authority as examples of regions with a broad range of demographics, shows dense innercity clusters with adjacent OAs of different types more common away from the centre. Additionally, it is clear that differences exist between regions; classification of areas surrounding the centre of Manchester more closely aligned with South East London than the rest of the city.

## 6 CONCLUSION AND FUTURE WORK

In this study, we proposed a novel framework for geodemographic analysis using supervised contrastive learning. By leveraging both the 2011 and 2021 Census datasets from England and Wales, we constructed graphs representing Output Areas and applied supervised contrastive learning to ob-

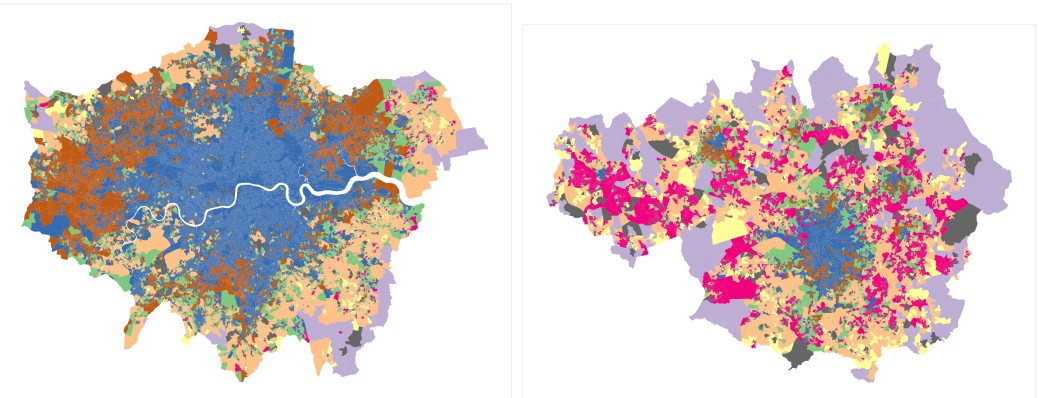

Figure 4: The confusion matrix showing coincidence of clusters between K-Means and BGMM. Only 53% of cluster assignments are shared, with methods identifying different aspects of the data.

Figure 5: The distribution of OA2011 clusters across Greater London (left) and Greater Manchester Combined Authority (right) using BGMM. Plots suggest central London geodemographics are distinct from the rest of the city, and East is different to West. Manchester also has a clear centre, with regions of differing demographics mixed throughout the rest of the combined authority area.

tain meaningful embeddings that capture the socio-economic evolution of regions. Our framework effectively combines historical data with attribute changes to infer the socio-economic conditions of Output Areas in 2021, offering a powerful tool for policymakers and stakeholders to review and simulate population dynamics over time.

We demonstrated the effectiveness of our approach through two classification tasks, where the 2021 Output Area embeddings were used to predict Output Area Classifications. Additionally, we explored clustering methods using Bayesian Gaussian Mixture Models to re-categorise Output Areas based on their learned 2021 embeddings.

Our findings suggest that graph-based approaches, particularly those incorporating SCL, offer significant advantages in achieving relatively balanced performance on imbalanced data while capturing complex patterns in geodemographic data. These approaches have the potential to be applied across various domains, including urban planning, public policy, and socio-economic analysis. Future work could explore extending this framework to other regions or integrating additional data sources, such as social media or environmental data, to further enhance the robustness and applicability of geodemographic models.

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
