# OpenReview forum: "Graph Supervised Contrastive Learning for Geodemographics"
_ICLR.cc/2025/Conference — ICLR 2025 Conference Withdrawn Submission_

### Official Review · Reviewer_up2D · 2024-11-03

**Soundness:** 2
**Presentation:** 3
**Contribution:** 2
**Rating:** 3
**Confidence:** 4

**Summary:**

This paper presents a new framework based on graph neural networks (GNNs) for modeling and analyzing temporal changes in demographic data. The proposed method is particularly designed to focus on socio-economic disparities across regions in England and Wales. The proposed framework adopts the Supervised contrastive learning strategy to learn Output Area (OA) embeddings, which can potentially improve the representation of minority classes. The effectiveness of the proposed framework is validated by two downstream tasks. The experimental results show that the proposed method can provide valuable insights for policymakers by better assessing the socio-economic transitions.

**Strengths:**

1. The problem tackled in this paper, i.e., modeling and analyzing the temporal changes in demographic data, is significant.
2. Graph neural networks demonstrate strong capabilities in the problem tackled in this paper, which may further extend the applicability of GNNs.

**Weaknesses:**

1. The proposed Supervised contrastive learning loss is quite similar to classical ones (e.g., ones in arxiv.org/pdf/2011.00362). It is not particularly tailored to the problem tackled in this paper. Authors are suggested to explain more on how the proposed Supervised contrastive learning strategy is different from classical ones and what unique aspects are considered by the contrastive learning strategy in this paper.
2. The methodological part of this paper is too simple. Authors should provide more technical details to help readers understand the proposed methodology. The authors are suggested to use some maths to interpret how the representations that are fed to the loss function are learned layer by layer.
3. The proposed framework does not perform well in some test cases.
4. The practical significance of the predictive results obtained by the proposed approach is not well discussed or analyzed. What can be discovered by the proposed approach and why the discovery is significant should be clearly discussed in the manuscript. For example, what latent patterns representing some important socio-economic transitions or disparities can be discovered by the proposed framework while not being discovered by existing approaches?
5. More ablation studies that consider different GNNs can be conducted.
6. Detailed settings of all approaches should be introduced to improve the reproducibility of the paper. Authors can refer to classical papers, e.g., the one introducing GATs, to describe how the network architecture and layer are configured in the experiments.

**Questions:**

1. Why is Supervised contrastive learning loss (Eq. (1)) adopted by the proposed framework? Are there any motivations behind this?
2. Why does the proposed framework not perform well on some test cases (e.g., Table 1)? Has the proposed framework been comprehensively fine-tuned in the experiments?
3. The practical significance of the predictive results obtained by the proposed approach should be comprehensively discussed.
4. More ablation studies that consider different GNNs can be conducted.
5. Are the results presented in the manuscript reproducible?

**Details Of Ethics Concerns:**

This reviewer has no critical ethical concerns.

---

### Official Review · Reviewer_AFBf · 2024-11-04

**Soundness:** 2
**Presentation:** 3
**Contribution:** 2
**Rating:** 3
**Confidence:** 4

**Summary:**

This paper proposes a framework for analysing and making predictions on geodemographics data.  More specifically, geographical regions are treated as nodes in a graph, with demographics data as node attributes. Embedding for the regions are learned through graph-based contrastive learning and used for downstream tasks such as classification and clustering of regions. Empirical results using real-world data in the UK were presented to illustrative the effectiveness of the proposed framework.

**Strengths:**

1) The paper addresses an interesting real-world application that is under-explored in the graph machine learning community.

2) The proposed framework is relatively clearly presented, with helpful visualisations to aid understanding.

3) Some explanation and discussion on processing real-world geodemographics data provide helpful guidelines for other researchers wishing to explore similar applications.

**Weaknesses:**

1) From a technical perspective the theoretical novelty of the work is perhaps limited for a conference like ICLR. This would be a very good submission to a multidisciplinary venue for example.

2) Some modelling choices are not clearly motivated. Why SCL? Why choosing GT and GraphSage? It would be good to provide more justifications. It is also unclear in SCL+NN what is the NN being used (is it a CNN?).

3) The empirical tasks are not very clearly explained. It seems OAC labels are being predicted but at the same time they are input to Algorithm 1. In addition, the empirical performance of the proposed SCL+NN approach does not seem encouraging enough against simple baselines, and the explanation for its poor performance is not entirely convincing.

**Questions:**

1) The colouring of the curvature in Fig.1 seems opposite to what is expected, with many central edges having negative curvature while some boundary ones having positive curvature. Can the authors clarify?

2) Giving that the geographical graph in Fig.1 looks planar and with relatively homogeneous node degrees, I am not sure if there are clear bottlenecks that would lead to over-squashing. It is also unclear why SCL would necessarily help with either over-smoothing or over-squashing.

3) Clustering and classification are typically treated as different problems. The paper somewhat mixed these two (eg in 2.2) which makes reading confusing at times.

4) The difference between SCL+GNN and GNN is not very clear (at least according to the description in 4.2).

---

### Official Review · Reviewer_XjQR · 2024-11-05

**Soundness:** 2
**Presentation:** 2
**Contribution:** 2
**Rating:** 5
**Confidence:** 3

**Summary:**

The paper focuses on using GNNs and Supervised Contrastive Learning for geodemographic analysis, with applications to the socio-economic evolution of regions in England and Wales between the 2011 and 2021 census periods. The framework builds embeddings for Output Areas by incorporating both historical data from 2011 and observed changes by 2021. Experimental results show the approach's effectiveness in classification and clustering tasks, particularly for minority classes.

**Strengths:**

1. The paper addresses a valuable problem by leveraging GNNs to analyze geodemographic data, capturing spatial relationships and socio-economic shifts over time. This type of analysis is crucial for understanding regional disparities and informing policy decisions.

2. The proposed framework, which combines SCL with GNNs, is well-suited to the application. The extensive experiments show the model's effectiveness, particularly in handling data from minority socio-economic groups, which are often challenging to represent accurately in such studies.

**Weaknesses:**

1. The paper could benefit from clearer explanations, especially on technical points. For instance:
a) It is not fully explained how the combined data from 2011 and 2021 is integrated into the second stage of SCL.
b) The approach for defining and using changes in Output Area attributes could be elaborated on.
c) Justification for the choice of the 2011 dataset to predict 2021 outcomes is unclear—why is 2011 specifically a suitable baseline?
d) The term "GT" in Table 1 is not defined, making it difficult for readers to interpret the results.

2.  While the application is innovative, the core method of using graph-based SCL is based on existing approaches rather than introducing a significant technical novelty. This may limit the perceived originality of the work, as it primarily adapts known techniques for a new application.

3. The experimental section would benefit from a comparison with non-GNN baselines, such as clustering algorithms, to highlight the unique advantages of the GNN approach. Including these baselines would make the results more compelling by providing a clearer perspective on the model's specific strengths and limitations relative to other methods.

4. No code provided

**Questions:**

Please refer to the weaknesses.

---

### Official Review · Reviewer_qVub · 2024-11-06

**Soundness:** 2
**Presentation:** 2
**Contribution:** 1
**Rating:** 3
**Confidence:** 4

**Summary:**

This paper presents a framework that combines Graph Neural Networks (GNNs) and Supervised Contrastive Learning (SCL) to analyze socio-economic changes between 2011 and 2021 in England and Wales, using geodemographic data. By leveraging SCL to enhance classification accuracy for minority groups and using temporal data to observe socio-economic transitions, the model provides embeddings that policymakers could use to track and simulate regional socio-economic shifts.

**Strengths:**

This paper applys GNNs in combination with SCL to address class imbalance in geodemographic data, which is a relatively unexplored approach in the field.

This paper leverages real-world census data from 2011 and 2021 for England and Wales.

This paper aims to improve classification outcomes for minority classes in geodemographic categories, which is essential given the data imbalances in this domain.

**Weaknesses:**

Lack of novelty: The proposed framework largely combines existing techniques (graph neural networks, supervised contrastive learning) without introducing truly novel methodological contributions. The application to geodemographics, while interesting, is not sufficiently innovative on its own.

Insufficient empirical evaluation: The experimental results are limited in scope and do not convincingly demonstrate the advantages of the proposed approach. Only a single dataset is used, and comparisons to relevant baselines are lacking.

Lack of baselines. The paper did not consider graph contrastive learning methods, including but not limited to the following:
Graph contrastive learning with augmentations
Graph contrastive learning automated
Infogcl: Information-aware graph contrastive learning
Deep graph contrastive representation learning

The paper could also benefit from more detailed analysis and examples of how the results and the proposed method translate into actionable insights in geodemographic contexts.

**Questions:**

see above

---

### Note · Authors · 2025-01-22

I have read and agree with the venue's withdrawal policy on behalf of myself and my co-authors.